# Deletions in *CWH43* cause idiopathic normal pressure hydrocephalus

Hong Wei Yang[1,2,3], Semin Lee[2,3], Dejun Yang[1], Huijun Dai[1], Yan Zhang[1], Lei Han[2,3], Sijun Zhao[1], Shuo Zhang[1], Yan Ma[1], Marciana F Johnson[2,3], Anna K Rattray[2,3], Tatyana A Johnson[2,3], George Wang[1,2,3], Shaokuan Zheng[1], Rona S Carroll[1,2,3], Peter J Park[2,3] & Mark D Johnson[1,2,3,4,*]

## Abstract

**Idiopathic normal pressure hydrocephalus (iNPH) is a neurological disorder that occurs in about 1% of individuals over age 60 and is characterized by enlarged cerebral ventricles, gait difficulty, incontinence, and cognitive decline. The cause and pathophysiology of iNPH are largely unknown. We performed whole exome sequencing of DNA obtained from 53 unrelated iNPH patients. Two recurrent heterozygous loss of function deletions in *CWH43* were observed in 15% of iNPH patients and were significantly enriched 6.6-fold and 2.7-fold, respectively, when compared to the general population. Cwh43 modifies the lipid anchor of glycosylphosphatidylinositol-anchored proteins. Mice heterozygous for *CWH43* deletion appeared grossly normal but displayed hydrocephalus, gait and balance abnormalities, decreased numbers of ependymal cilia, and decreased localization of glycosylphosphatidylinositol-anchored proteins to the apical surfaces of choroid plexus and ependymal cells. Our findings provide novel mechanistic insights into the origins of iNPH and demonstrate that it represents a distinct disease entity.**

**Keywords** CWH43; GPI-anchored protein; hydrocephalus; normal pressure hydrocephalus
**Subject Categories** Genetics, Gene Therapy & Genetic Disease; Neuroscience
See also: **V Leinone et al** (March 2021)

## Introduction

Idiopathic normal pressure hydrocephalus (iNPH) is a neurological disorder of aging that is characterized by enlarged cerebral ventricles, gait difficulty, incontinence, and cognitive impairment (Adams *et al*, 1965). Because the symptoms can be improved by CSF drainage (Adams *et al*, 1965; Walchenbach *et al*, 2002; Wilson &

Williams, 2010), iNPH is classified as a reversible dementia. According to the Hydrocephalus Association, iNPH affects about 700,000 Americans. It occurs almost exclusively after the age of 60 (the average age at diagnosis is about 75 years). Researchers have estimated that 1.4–2.9% of the population over the age of 65 (Hiraoka *et al*, 2008; Jaraj *et al*, 2014), 5.6% of the population over the age of 75 (Hiraoka *et al*, 2008; Martin-Laez *et al*, 2015), and nearly 1 in 7 nursing home residents (Marmarou *et al*, 2007) has iNPH. Unfortunately, most of these patients are misdiagnosed or undiagnosed (Marmarou *et al*, 2007; Hiraoka *et al*, 2008; Martin-Laez *et al*, 2015), in part because the symptoms of iNPH resemble the symptoms of other neurological disorders such as Parkinson's Disease or Alzheimer's Disease (Jaraj *et al*, 2014), and in part because many healthcare providers and the lay public are unfamiliar with this disorder (Conn & Lobo, 2008). Importantly, not all patients who present with ventriculomegaly, gait difficulty, incontinence, and cognitive decline improve after CSF drainage (Relkin *et al*, 2005; McGirt *et al*, 2008), such that a diagnosis of iNPH is only confirmed if the symptoms improve after CSF drainage and shunt placement (Relkin *et al*, 2005; McGirt *et al*, 2008). This empirical approach to diagnosis, combined with the similarity to other disorders and the lack of knowledge about its pathophysiology, has led some practitioners to question whether iNPH exists (Saper, 2017).

Although first described in 1965 (Adams *et al*, 1965), the etiology and pathophysiology of iNPH remain a mystery. Ventricular CSF stasis (Kawaguchi *et al*, 2011), abnormal cerebrovascular blood flow and reactivity (Kristensen *et al*, 1996; Mori *et al*, 2002), and increased amplitude of intracranial pressure waves (Eide & Sorteberg, 2010) have all been observed in iNPH patients, but the mechanisms underlying these phenomena are unclear. Associations between iNPH and hypertension, hypercholesterolemia, diabetes, and alcohol consumption have been reported (Jacobs, 1977; Krauss *et al*, 1996; Eide & Pripp, 2014; Hickman *et al*, 2017), but the physiological mechanisms underlying these associations are not known. Several reports of familial iNPH have been published, including pedigrees with autosomal-dominant transmission (Zhang *et al*, 2008; Takahashi *et al*, 2011; McGirr & Cusimano, 2012; Huovinen *et al*, 2016; Morimoto *et al*,

1 University of Massachusetts Medical School, Worcester, MA, USA
2 Brigham and Women's Hospital, Boston, MA, USA
3 Harvard Medical School, Boston, MA, USA
4 UMass Memorial Health Care, Worcester, MA, USA
*Corresponding author. Tel: +1 508 334 0605; Fax: +1 508 856 5074; E-mail: mark.johnson3@umassmemorial.org

2019). One study identified a *CFAP43* mutation as a possible cause of familial iNPH (Morimoto *et al*, 2019). A recent single-nucleotide polymorphism study reported intronic copy number loss in the *SFMBT1* gene in 26% of sporadic iNPH patients compared to 4.2% of the general population (Sato *et al*, 2016). Another large multinational study of Finnish and Norwegian iNPH patients used genotyping of *SFMBT1* to identify copy number loss in intron two of *SFMBT1* in 10% of Finnish [odds ratio (OR) = 1.9, $P = 0.0078$] and 21% of Norwegian (OR = 4.7, $P < 0.0001$) patients with iNPH (Korhonen *et al*, 2018), although the significance of this finding is yet to be determined. The current study was designed to identify genetic abnormalities associated with shunt-responsive iNPH.

# Results

## Patient characteristics

Whole exome sequencing of DNA obtained from 53 patients with shunt-responsive iNPH was performed in three independent cohorts (Table 1). Collectively, there were 29 females and 24 males. The median age was 75 years (range 65–89 years). All of the patients had enlarged cerebral ventricles and gait difficulty. Urinary incontinence and cognitive impairment were present in 79 and 83% of the patients, respectively.

## Recurrent iNPH-associated deletions

Analysis of sequencing data identified 4 of 53 patients with the same damaging mutation in *CWH43* (Appendix Fig S1). The presence of the mutation was confirmed in each case by Sanger sequencing (Fig 1A). The mutation (4:49063892 CA/C; Lys696AsnfsTer23) has a minor allele frequency (MAF) of 0.0057 in the general population and 0.0377 among iNPH patients, a 6.6-fold increase ($P < 0.0001$, $X^2$ test; $P < 0.0002$, $X^2$ test with Yates correction). This deletion, which was heterozygous in each patient and observed in 2 of 3 independent cohorts, leads to a frameshift that alters the carboxyl terminus of Cwh43.

Manual examination of sequencing data identified 4 additional patients harboring a different *CWH43* deletion (4:49034669 CA/C; Leu533Ter) (Appendix Fig S2). This *CWH43* deletion generates a frameshift that causes premature termination of the Cwh43 protein (Fig 1A). This second deletion was filtered out during the original analysis because its MAF exceeds 0.01 in the general population. This *CWH43* deletion has a MAF of 0.0142 in the general population and 0.0377 in our iNPH cohort, and thus occurred with a 2.7-fold increased frequency ($P < 0.0406$, $X^2$ test; $P < 0.1016$, $X^2$ test with Yates correction). This second *CWH43* deletion was also heterozygous in each patient and observed in 2 of 3 independent cohorts.

Taken together, 8 of 53 iNPH patients (15%) carried a recurrent *CWH43* deletion that disrupts the carboxyl terminus of the Cwh43 protein. Each of the eight patients presented with gait difficulty, incontinence, and cognitive impairment that improved after CSF drainage. Three of these patients had a family history of iNPH or gait difficulty, with one patient describing three first degree relatives who had been diagnosed with iNPH.

Axial T2-weighted MR images of the brain were obtained from the eight iNPH patients harboring *CWH43* deletions and 54 asymptomatic individuals matched for age and gender (www.oasis-brains.org). The ratio of ventricular to brain cross-sectional area for each case was obtained. The mean ratio (representing the normalized ventricular size) among the 8 iNPH patients with *CWH43* deletions (21.7 ± 1.4) was significantly larger than that of the cohort of 54 asymptomatic age- and gender-matched controls (11.3 ± 0.4; $P < 0.00001$, two-tailed $t$-test; Appendix Fig S3).

## Effect of CWH43 deletions on Cwh43 function

Yeast Cwh43 (Cell wall biogenesis protein 43 C-terminal homolog) is a transmembrane protein that incorporates ceramide into the glycosylphosphatidylinositol (GPI) anchor that attaches certain proteins to the cell membrane (Ghugtyal *et al*, 2007). Incorporation of ceramide into the lipid anchor of GPI-anchored proteins by Cwh43 regulates their membrane localization in yeast (Yoko *et al*, 2018). Although the function of Cwh43 in multicellular organisms is not known, a lipid-remodeling domain is predicted to reside near the carboxyl terminus (Fig 1B). The recurrent iNPH-associated *CWH43* mutation (Lys696AsnfsTer23) causes a frameshift that eliminates the last four amino acids of Cwh43 and replaces it with a novel 23 amino acid sequence. This removes an endoplasmic reticulum (ER) export signal (YF) and adds an ER retrieval signal (KKXX, Fig 1B) (Matheson *et al*, 2006). Overexpression of wild-type GFP-Cwh43 or mutant GFP-Cwh43-k696fs fusion protein in HeLa cells confirmed that wild-type GFP-Cwh43 is associated with the ER, intracellular vesicles, and the Golgi apparatus (where modification of the lipid anchor is thought to occur), while GFP-Cwh43-k696fs is localized primarily to the ER (Fig 1C).

The other recurrent *CWH43* deletion creates a stop codon at Leu533, thereby generating a truncated Cwh43 protein lacking the C-terminal ER export signal and the putative lipid-remodeling domain. Overexpression of human GFP-Cwh43-Leu533Ter in HeLa cells confirmed that the mutant protein is diffusely distributed throughout the cytoplasm (Fig 1C).

We generated two independent HeLa cell lines containing a mutation that truncates the Cwh43 protein at or near Leu533. Western blot analysis indicated that Cwh43 protein was essentially undetectable in these cells (Appendix Fig S4). Transient expression of RFP fused folate receptor alpha and CD59, two GPI-anchored proteins, showed that loss of Cwh43 expression decreases the association of GPI-anchored proteins with intracellular vesicles (Appendix Fig S4). However, flow cytometry indicated that the amount of CD59 on the surface of *CWH43* mutant HeLa cell lines was not decreased (Appendix Fig S4). Subfractionation of wild-type and *CWH43* mutant HeLa cells into aqueous and lipid compartments using Triton X-114, followed by Western blot analysis, revealed that the Cwh43 Leu533Ter deletion decreases the association of CD59 with the lipid microdomain fraction where GPI-anchored proteins are typically found (Ko & Thompson, 1995) (Fig 1D), even though the amount of CD59 in the total membrane fraction increased slightly. The effect of Cwh43 mutation on the localization of CD59 to the lipid microdomain fraction could be rescued by overexpression of wild-type GFP-Cwh43, but not by GFP-Cwh43-k696fs mutant protein (Fig 1D). These findings suggest that Cwh43 regulates the membrane targeting of GPI-anchored proteins in human cells. Both of the iNPH-associated *CWH43* deletions disrupt this function.

**Table 1.  iNPH patient characteristics.**

| Cohort | Age | Sex | Sx duration | Gait | Incontinence | Dementia | Improvement | *CWH43* alteration |
|---|---|---|---|---|---|---|---|---|
| I | 76 | F | NA | ** | *** | ** | ** | K669fs |
| I | 79 | M | NA | ** | * | None | * | |
| I | 84 | F | 2 | *** | ** | *** | * | |
| I | 80 | F | 2 | ** | ** | *** | * | |
| I | 80 | F | 5 | *** | ** | ** | *** | Leu533Ter |
| I | 68 | M | NA | *** | None | None | ** | |
| I | 76 | F | 2 | ** | * | *** | * | |
| I | 74 | M | 4 | *** | * | * | *** | K669fs |
| I | 84 | F | 2 | *** | ** | None | ** | |
| I | 77 | M | 6 | ** | None | * | * | |
| I | 65 | F | 4 | *** | *** | ** | *** | K669fs |
| I | 76 | F | 2 | * | * | * | ** | |
| I | 69 | M | 2 | *** | * | ** | ** | Leu533Ter |
| I | 67 | M | 1 | ** | ** | ** | *** | |
| I | 76 | M | NA | ** | *** | ** | ** | |
| I | 77 | F | 2 | *** | ** | ** | ** | |
| I | 70 | F | 2 | ** | ** | ** | ** | |
| I | 68 | M | 3 | *** | ** | ** | *** | Leu533Ter |
| I | 89 | M | 10 | *** | *** | ** | *** | |
| I | 81 | M | 1 | *** | *** | ** | *** | |
| II | 75 | M | 2 | * | ** | * | ** | |
| II | 76 | F | 2 | *** | ** | ** | *** | |
| II | 81 | F | 1 | *** | *** | ** | *** | |
| II | 81 | F | 1 | *** | ** | ** | ** | |
| II | 86 | M | 3 | *** | ** | ** | ** | |
| II | 75 | F | NA | *** | ** | ** | * | |
| II | 81 | F | 1 | *** | None | * | * | |
| II | 76 | M | 2 | *** | ** | ** | * | |
| II | 84 | F | 3 | *** | * | * | * | K669fs |
| II | 73 | F | 4 | *** | *** | ** | ** | |
| II | 70 | M | 2 | ** | *** | *** | * | |
| II | 75 | F | 2 | ** | ** | ** | ** | |
| III | 77 | M | 20 | ** | None | None | ** | |
| III | 72 | F | 0.5 | ** | *** | *** | * | |
| III | 78 | M | 1 | ** | * | * | ** | |
| III | 73 | M | NA | ** | None | None | ** | |
| III | 70 | F | 2 | ** | * | * | *** | |
| III | 78 | M | 2 | ** | None | * | ** | |
| III | 75 | F | 0.5 | ** | ** | * | ** | |
| III | 68 | F | 3 | ** | ** | * | * | |
| III | 77 | M | 3 | *** | None | None | ** | |
| III | 75 | M | 2 | ** | None | ** | ** | |
| III | 70 | F | 2 | ** | ** | None | *** | |
| III | 68 | M | 2 | ** | ** | ** | *** | |
| III | 76 | M | NA | *** | *** | ** | *** | |

**Table 1** (continued)

| Cohort | Age | Sex | Sx duration | Gait | Incontinence | Dementia | Improvement | *CWH43* alteration |
|--------|-----|-----|-------------|------|--------------|----------|-------------|---------------------|
| III | 75 | F | 2 | *** | ** | ** | ** | |
| III | 68 | M | 1 | ** | ** | ** | ** | Leu533Ter |
| III | 72 | F | 1 | * | * | None | *** | |
| III | 72 | F | 1 | * | None | None | * | |
| III | 81 | F | NA | *** | ** | *** | ** | |
| III | 74 | M | 2 | *** | None | ** | *** | |
| III | 79 | F | 1 | *** | *** | *** | *** | |
| III | 70 | F | 0.5 | *** | * | * | * | |

NA, Data not available; None, Symptom not present.
*Mild, **Moderate, and ***Major.

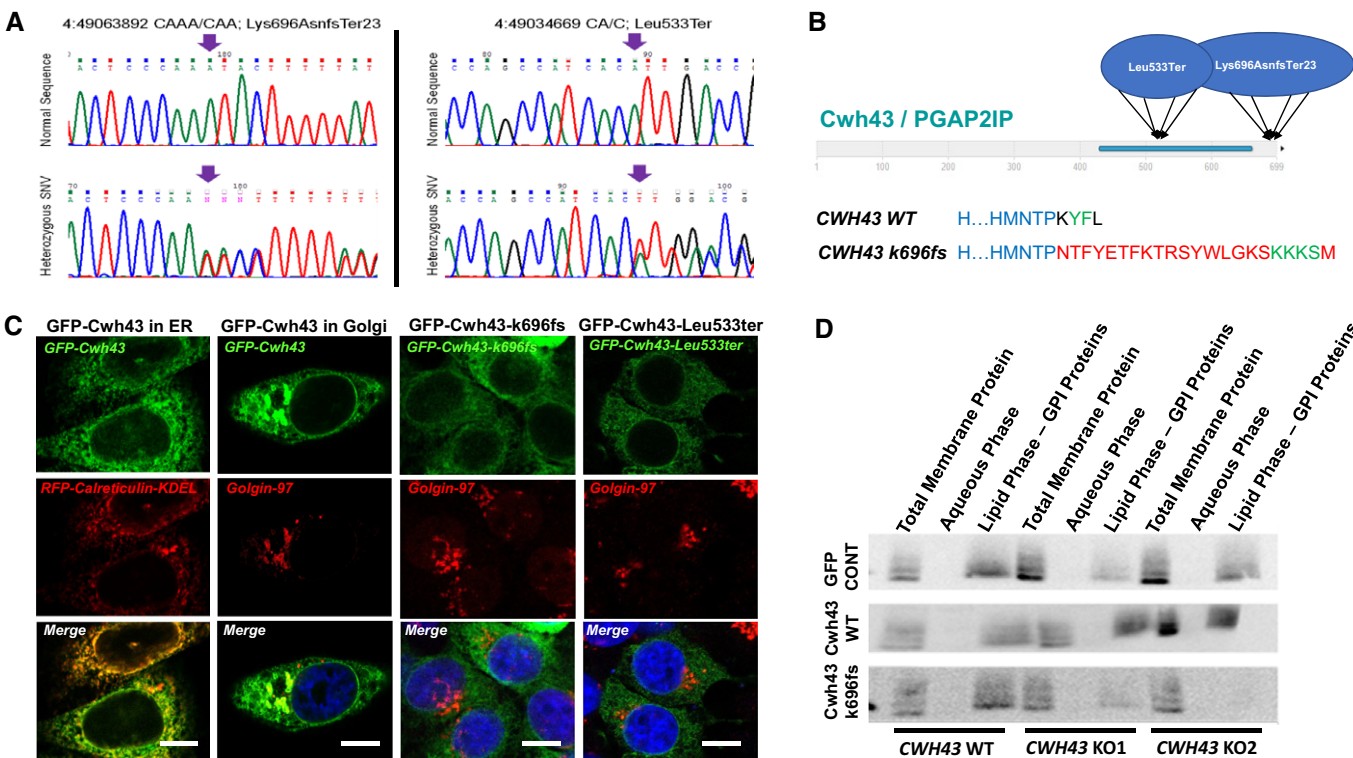

**Figure 1. iNPH-associated deletions disrupt the ability of Cwh43 to regulate the localization of GPI-anchored receptors.**

A    Sanger DNA sequencing data from two iNPH patients confirming the presence of heterozygous *CWH43* deletions. Arrows identify location of the deletion.

B    Diagram of the domain structure of the Cwh43 protein illustrating the location of the damaging *CWH43* deletions. Lower panel details the disruptive effect of the k696Asnfs mutation on the C-terminal endoplasmic reticulum export signal (YF), replacing it via a frameshift with a 23 amino acid sequence that contains an endoplasmic reticulum retention signal (KKKS).

C    Fluorescence micrographs of HeLa cells transfected with plasmids encoding human GFP-Cwh43, GFP-Cwh43-k696fs, or GFP-Cwh43-Leu533ter fusion proteins. To label the ER, HeLa cells were transduced with a baculovirus encoding a fluorescently labeled RFP-calreticulin fusion protein containing an ER retention signal (RFP-calreticulin-KDEL). To label the Golgi apparatus, cells were immunostained for golgin-97 (red). When compared with wild-type GFP-Cwh43, GFP-Cwh43-k696fs, and GFP-Cwh43-Leu533 showed decreased association with the Golgi apparatus (but not the ER) and were diffusely distributed throughout the cytoplasm. Scale bar is approximately 5 μm.

D    Western blot analysis of total membrane, aqueous and lipid (GPI-anchor-containing) Triton X-114 extracts derived from wild-type HeLa cells, and two independent CRISPR *CWH43* knockout (KO) HeLa cell lines in which a mutated *CWH43* gene encodes a protein that is truncated near Leu533 and CWH43 mRNA and protein are markedly reduced. Cells were transfected to overexpress a control GFP plasmid, a plasmid encoding human wild-type Cwh43 with GFP fused to the N-terminus, or a plasmid encoding human *CWH43* harboring the iNPH-associated mutation (Lys696AsnfsTer23) with GFP fused to the N-terminus. The Western blot was stained using an antibody directed against CD59, a GPI-anchored protein.

Source data are available online for this figure.

### Cwh43 expression in the mouse brain

Mouse mRNA *in situ* hybridization images (Lein *et al*, 2007) revealed increased Cwh43 mRNA expression in the choroid plexus, layer CA1-CA3 of the hippocampus, several thalamic nuclei, and layer V of the cerebral cortex (Fig 2A). Using frozen cryostat sections of the mouse brain, we found that Cwh43 immunoreactivity was concentrated in the ventricular ependymal layer (Fig 2B) and choroid plexus (Appendix Fig S5). In cultured mouse ependymal cells, Cwh43 immunoreactivity was observed in the soma and in motile cilia (Fig 2C).

### Effect of Cwh43 deletion in mice

Using CRISPR/Cas9 technology, we generated two independent lines of *CWH43* mutant mice (Appendix Fig S6). *CWH43^{M533}* mice harbor a mutation (Met533Ter) corresponding to human 4:49034669 CA/C; Leu533Ter. We generated a second mouse line (*CWH43^{M533/A530}*) harboring one *CWH43^{M533}* allele and a *CWH43* allele that results in termination of Cwh43 at A530, three amino acids before Met533. Heterozygous *CWH43^{WT/M533}*, homozygous *CWH43^{M533/M533}*, and compound heterozygous *CWH43^{M533/A530}* mice appeared grossly normal and were fertile.

We used MRI to assess ventricular volume in the brains of 6-month-old wild-type and *CWH43* mutant mice (Fig 3A and B). When compared to *CWH43^{WT/WT}* mice, ventricular volume was increased by approximately 24.2% in *CWH43^{WT/M533}* heterozygous mice (*P* < 0.0015, *n* = 8, unpaired *t*-test), 18.3% in *CWH43^{M533/M533}* homozygous mice (*P* < 0.0014, *n* = 5, unpaired *t*-test), and 20.8% in *CWH43^{M533/A530}* compound heterozygous mice (*P* < 0.0064, *n* = 8, unpaired *t*-test). Light microscopy indicated that the brains of *CWH43* mutant mice were grossly normal and the cerebral aqueduct was patent. Injection of fluorescent dextran (70 kDa) into the lateral ventricle resulted in filling of the fourth ventricle with dextran within 10 min, confirming that this was a communicating hydrocephalus (Appendix Fig S7). Immunohistochemistry for Cwh43 revealed a loss of Cwh43 immunoreactivity in the ventricular epithelia of *CWH43^{M533/M533}* mice (Appendix Fig S8).

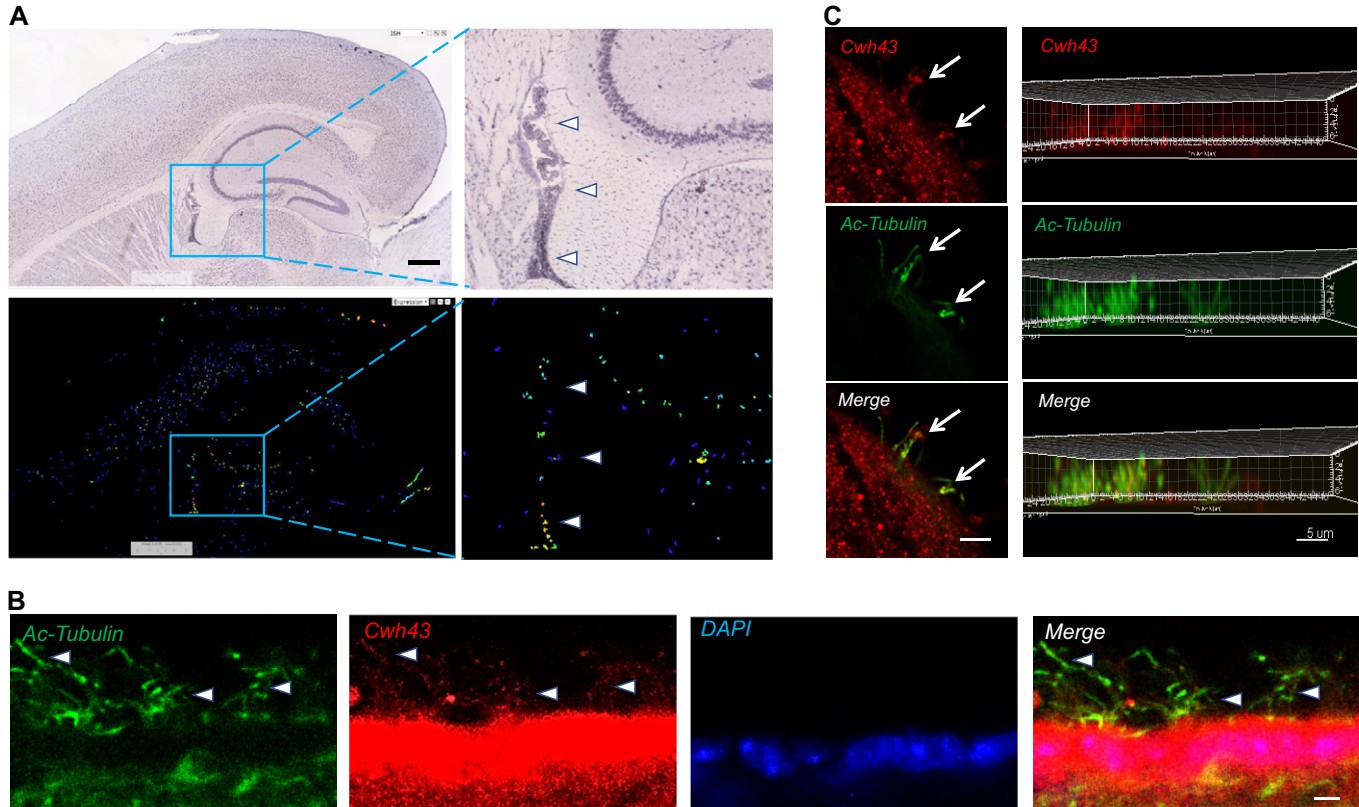

**Figure 2. Expression of Cwh43 mRNA and protein in the mouse brain.**

A  mRNA *in situ* hybridization images showing expression of Cwh43 mRNA in the mouse brain. Enclosed areas containing potions of the ventricle, hippocampus, and dorsal thalamus are shown at higher magnification on the right. Arrowheads point to choroid plexus. Scale bar is approximately 400 μm.

B  Fluorescence immunohistochemistry of the ependymal surface of the lateral ventricle of the mouse brain. Cilia are visualized using an antibody for acetylated alpha tubulin (green). Cwh43 is visualized using a specific anti-Cwh43 antibody (red). Nuclei are counterstained using DAPI (blue). Arrowheads point to motile cilia and scale bar is approximately 5 μm.

C  Confocal fluorescence immunocytochemistry images of a single cultured mouse ciliated ependymal cell. Cilia were visualized using an antibody for acetylated alpha tubulin (green). Cwh43 immunoreactivity was visualized using a specific anti-Cwh43 antibody (red). Scale bar (left column) is approximately 4 μm. Images in the column on the right represent a Z-stack reconstruction of confocal images showing localization of Cwh43 immunoreactivity in cilia of a mouse ependymal cell. Scale bar (right column) is approximately 5 μm.

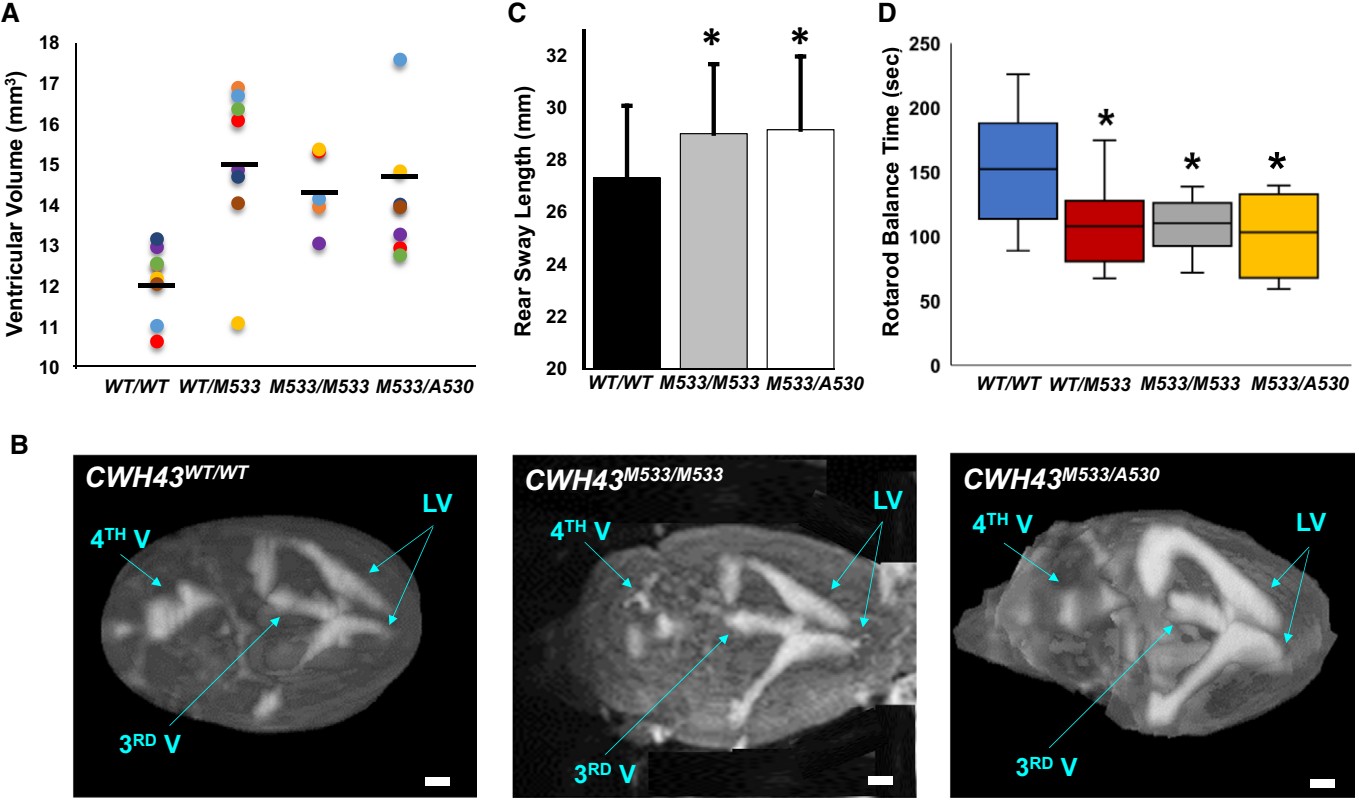

**Figure 3. *CWH43* mutation increases ventricular volume and causes gait dysfunction in mice.**

A   Scatter plot comparing ventricular volume from *CWH43*^WT/WT^ (wild-type, WT), heterozygous *CWH43*^WT/M533^, homozygous *CWH43*^M533/M533^, and *CWH43*^M533/A530^ mice at 6 months. Ventricular volume was calculated from T2-weighted MR images of the mouse brain using a custom automated computer algorithm. Horizontal bars indicate the mean of the measurements in each column. Statistical significance for each mutant mouse line compared to WT was determined using the unpaired *t*-test ($P = 0.0015$ for *CWH43*^WT/M533^; $P = 0.0014$ for *CWH43*^M533/M533^; $P = 0.006$ for *CWH43*^M533/A530^).

B   Representative 3D volumetric MR images of mouse brains from 6-month-old *CWH43*^WT/WT^, *CWH43*^M533/M533^, and *CWH43*^M533/A530^ mice. LV = lateral ventricle, 3^rd^ V = third ventricle, 4^th^ V = fourth ventricle. Scale bar is approximately 1 mm.

C   Quantitative gait analysis at 7 months of age revealed increased sway among homozygous *CWH43*^M533/M533^ mice, (*$P = 0.03$, $n = 5$) and compound heterozygous *CWH43*^M533/A530^ (*$P = 0.034$, $n = 4$) mice when compared to wild-type *CWH43*^WT/WT^ mice ($n = 5$). Sway (the distance between the hind paws during walking) was measured repeatedly for individual mice in each group during a constrained unidirectional walk. Data shown are the mean ± SD for each group. Statistical significance was determined using the unpaired *t*-test.

D   Box plot showing rotarod performance data for *CWH43*^WT/WT^, *CWH43*^WT/M533^, *CWH43*^M533/M533^, and *CWH43*^M533/A530^ mice at 7 months of age. Data shown are the mean, 1^st^ quartile, 3^rd^ quartile, minimum, and maximum for each group of mice. Statistical significance was determined using the unpaired *t*-test. When compared to wild-type *CWH43*^WT/WT^ mice ($n = 10$), balance time on the rotarod was decreased significantly among heterozygous *CWH43*^WT/M533^ mice (*$P = 0.04$, $n = 8$), homozygous *CWH43*^M533/M533^ mice (*$P = 0.03$, $n = 7$), and compound heterozygous *CWH43*^M533/A530^ mice (*$P = 0.03$, $n = 4$).

Source data are available online for this figure.

Quantitative gait analysis in *CWH43* mutant mice at 7 months revealed increased rear leg sway (Fig 3C, $P < 0.011$, $n = 9$, unpaired *t*-test) when compared to wild-type mice. Using the rotarod performance test to evaluate balance and coordination, we observed significantly decreased balance times for *CWH43*^WT/M533^ heterozygous mice ($P < 0.0405$, $n = 8$, unpaired *t*-test), *CWH43*^M533/M533^ homozygous mice ($P < 0.0323$, $n = 7$, unpaired *t*-test), and *CWH43*^M533/A530^ compound heterozygous mice ($P < 0.03$, $n = 4$, unpaired *t*-test) when compared to wild-type mice (Fig 3D).

Electron microscopy examination of the mouse brain ventricular surface revealed a decrease in the number of ependymal cilia by approximately 28% in *CWH43*^M533/M533^ homozygous mice ($P < 0.0037$, $n = 4$, unpaired *t*-test, Fig 4A) and 25% in *CWH43*^M533/A530^ compound heterozygous mice ($P < 0.0003$, $n = 6$, unpaired *t*-test,

Appendix Fig S9) when compared to wild-type mice. The number of microvilli on the ependymal surface also appeared to be decreased in *CWH43* mutant mice. Obvious differences in cilia length were not observed. Immunohistochemistry confirmed the decrease in ependymal cilia in *CWH43*^M533/A530^ mice (Appendix Fig S9).

We used Triton X-114 to subfractionate brain and kidney tissues from *CWH43*^WT/WT^, *CWH43*^WT/M533^, and *CWH43*^M533/M533^ mice into aqueous and lipid compartments. Western blot analysis revealed that *CWH43* mutation decreases the association of the GPI-anchored protein, CD59, with the lipid microdomain fraction in heterozygous *CWH43*^WT/M533^ and homozygous *CWH43*^M533/M533^ mice *in vivo* (Fig 4B). Immunohistochemistry revealed that *CWH43* mutation redirected the localization of CD59 from the apical membrane to the

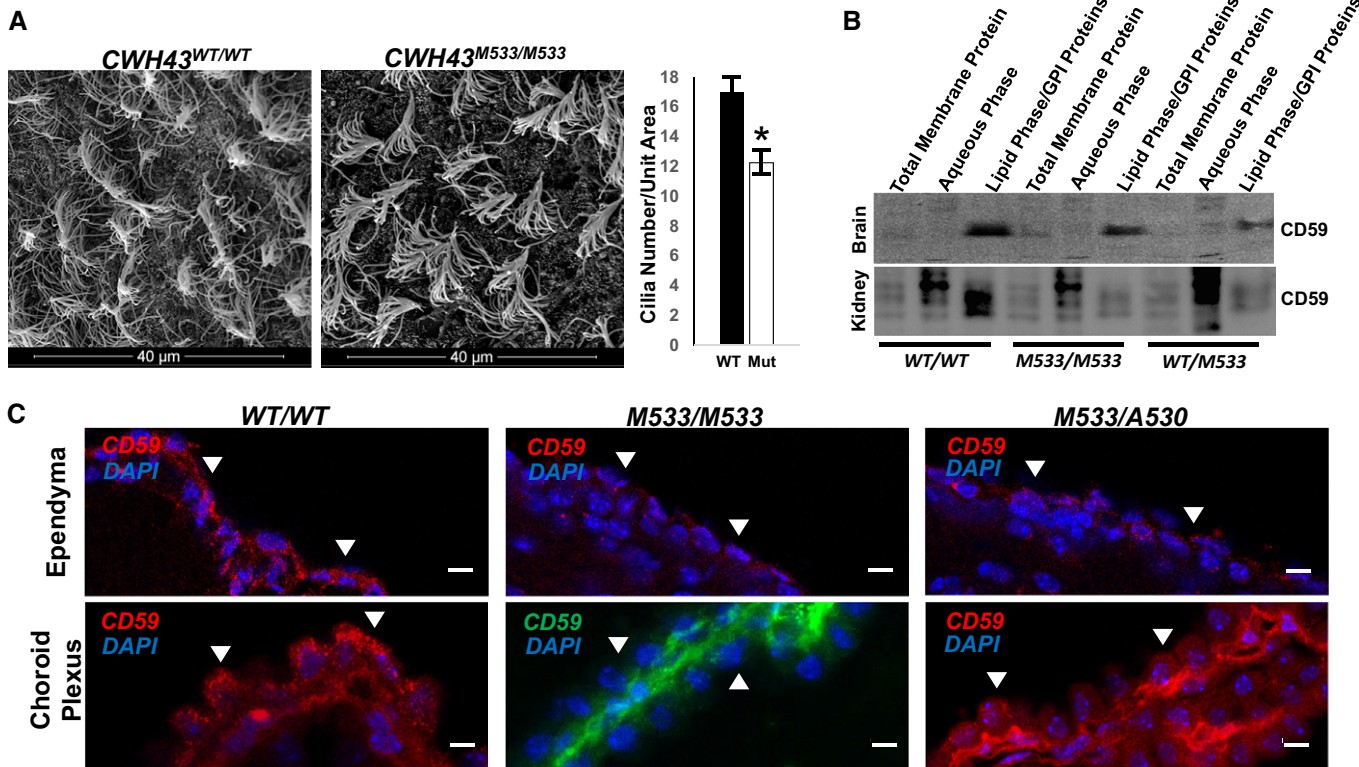

**Figure 4. *CWH43* mutation decreases cilia number and alters the distribution of GPI-anchored proteins in mouse ventricular ciliated epithelia.**

A   Scanning electron micrographs of the ependymal surface of the lateral ventricle of *CWH43* WT and *CWH43^M533/M533^* mice. Scale bar is approximately 40 μm. The graph on the right quantifies the data from the electron micrographs on the left. Data shown are the mean ± SEM. *P = 0.004, n = 4, unpaired *t*-test.

B   Western blot analysis of total membrane, aqueous, and lipid (GPI-anchor-containing) Triton X-114 extracts derived from wild-type, *CWH43^M533/M533^*, and *CWH43^WT/M533^* mouse brain or kidney. The Western blot was stained using an antibody directed against CD59, a GPI-anchored protein.

C   Fluorescence immunohistochemistry for CD59 in the ependymal layer and choroid plexus of the lateral ventricle from *CWH43^WT/WT^*, *CWH43^M533/M533^*, and *CWH43^M533/A530^* mice. Arrowheads point to apical surfaces of ependymal and choroid plexus cells. Nuclei are counterstained using DAPI (blue). Scale bar is approximately 5 μm.

Source data are available online for this figure.

basal membrane of multiciliated choroid plexus and ependymal cells in *CWH43^M533/M533^* and *CWH43^M533/A530^* mice (Fig 4C).

## Discussion

As the US population ages, iNPH has become a growing (albeit underappreciated) public health problem. iNPH patients often suffer from an inability to walk unassisted, frequent falls, urinary or fecal incontinence, and cognitive impairments that prevent them from living independently (Adams *et al*, 1965; McGirt *et al*, 2005; Graff-Radford, 2007). Nearly 6% of individuals will develop this disorder by the age of 75 (Hiraoka *et al*, 2008; Tanaka *et al*, 2009; Martin-Laez *et al*, 2015), and estimates suggest that 14% of nursing home residents have iNPH (Marmarou *et al*, 2007). Thus, our finding that iNPH-associated *CWH43* deletions occur in approximately 3% of the general population is in keeping with current prevalence estimates of iNPH (Marmarou *et al*, 2007; Tanaka *et al*, 2009; Martin-Laez *et al*, 2015). Although each of the three cohorts in our study was limited in size, we nevertheless observed that *CWH43* deletions were overrepresented in each one. Importantly, *CWH43* deletions

accounted for only 15% of iNPH patients in our cohort, suggesting that other genetic or environmental factors (Sato *et al*, 2016; Hickman *et al*, 2017) contribute to the development of iNPH.

Most iNPH patients do not report a family history of the disease (Huovinen *et al*, 2016), but autosomal-dominant transmission has been reported (Zhang *et al*, 2008; Takahashi *et al*, 2011; McGirr & Cusimano, 2012; Morimoto *et al*, 2019). Although a family history of iNPH was not required for entry into this study, three of eight patients with *CWH43* deletions reported a family history of iNPH or gait difficulty. Importantly, we observed that humans and mice heterozygous for iNPH-associated *CWH43* deletions display an increase in ventricular volume as well as gait and balance dysfunction, consistent with an autosomal-dominant pattern of inheritance.

Although iNPH patients appear functionally normal at birth and only develop symptoms after the sixth decade of life, they can develop progressive ventriculomegaly prior to symptom onset (Engel *et al*, 2018). Likewise, *CWH43* mutant mice appear normal at birth and show no major deficits through middle age. However, careful testing revealed enlarged ventricles and gait dysfunction during this period. This animal model of iNPH thus accurately reflects the phenotype and time course of the disease.

Mutations affecting proteins involved in GPI-anchored protein synthesis can cause mental retardation, microcephaly, and seizures, while complete disruption of GPI-anchored protein synthesis is lethal (Kinoshita, 2014). Although Cwh43 modifies the lipid anchor of certain GPI-anchored proteins, it is not required for their basic synthesis. This fact may help to explain the absence of symptoms until late in life in iNPH patients.

In yeast, the incorporation of ceramide into the lipid anchor of GPI-anchored proteins by Cwh43 promotes the retention of these proteins on the plasma membrane (Yoko *et al*, 2018). Loss of Cwh43 function in yeast leads to aberrant expression of these proteins on the cell wall. The Golgi apparatus is thought to be a primary site where the lipid anchor of GPI-anchored proteins is modified and where GPI-anchored proteins are incorporated into vesicles for transport to their final destination on the cell membrane (Matheson *et al*, 2006). We observed strong expression of Cwh43 in the Golgi apparatus of mammalian cells. However, both of the iNPH-associated *CWH43* deletions cause a loss of Cwh43 protein in the Golgi apparatus and disrupt the targeting of certain GPI-anchored proteins (e.g., CD59) to lipid microdomains in the plasma membrane and to the apical membrane of polarized ventricular epithelial cells. Our data thus provide the first evidence that Cwh43 regulates the subcellular targeting of GPI-anchored proteins in mammalian cells. We hypothesize that mislocalization of GPI-anchored proteins in choroid plexus and ependymal epithelial cells harboring mutant Cwh43 disrupts the normal function of these cells.

A decline in the number of ciliated ventricular cells has been observed in aged asymptomatic individuals with enlarged ventricles and in patients with chronic hydrocephalus (Del Bigio, 1993; Shook *et al*, 2014). Published reports regarding genetic causes of hydrocephalus in humans and in mice, as well as numerous experimental studies of induced hydrocephalus, have implicated loss or dysfunction of ventricular multiciliated epithelia in the etiology of many forms of hydrocephalus (Del Bigio, 1993; Huh *et al*, 2009; Shook *et al*, 2014; Narita & Takeda, 2015; Kahle *et al*, 2016; Morimoto *et al*, 2019). Our finding that *CWH43* mutations affect ependymal cilia number and apical/basal targeting of GPI-anchored proteins in ventricular multiciliated epithelial cells suggests that dysfunction of these cells may contribute to the development of iNPH. We propose a model in which decreased Cwh43 function disrupts the trafficking of GPI-anchored proteins and leads to a decrease in the function and/or number of ventricular multiciliated cells and ventricular enlargement. This combines with age-related declines in cell number to produce an age-dependent compromise of ventricular multiciliated cell function, ventricular enlargement, and iNPH onset at an advanced age. Further studies will examine whether *CWH43* mutations lead to hydrocephalus via additional mechanisms.

Many practitioners are unfamiliar with iNPH (Conn & Lobo, 2008), and some even question whether iNPH exists as a separate disease entity (Saper, 2017). This skepticism and lack of familiarity contribute to the low rates of diagnosis and treatment of this disorder. Our finding that loss of function *CWH43* deletions are enriched among iNPH patients and produce an iNPH-like syndrome in mice provides key mechanistic insights into the etiology of iNPH and firmly establish this disorder as a distinct disease.

# Materials and Methods

## Patients

Fifty-three unrelated patients who presented to a neurosurgical clinic with unexplained complaints of ventricular enlargement, gait difficulty, incontinence, and/or cognitive decline underwent an evaluation that included a history, neurological examination, and cranial imaging. Patients were consented for the study prior to CSF drainage. The experiments conformed to the principles set out in the WMA Declaration of Helsinki and the Department of Health and Human Services Belmont Report. A trial of lumbar CSF drainage was then performed. Quantitative measurements of gait speed, stride length, and performance on the Timed Up and Go (TUG) test, as well as patient and caregiver reports of changes in urinary incontinence, were used to assess symptoms before and after CSF drainage as we have described previously (Yang *et al*, 2016). Only those patients who improved after a trial of CSF drainage and subsequent ventriculoperitoneal shunt placement were included for whole exome analysis.

## Whole exome sequencing and data analysis

The 53 patients with shunt-responsive iNPH were enrolled and analyzed in three separate cohorts ($n = 20$, $n = 12$, and $n = 21$). Genomic DNA was isolated from whole blood and submitted for whole exome sequencing (50× coverage, 150 bp paired-end sequencing, Illumina HiSeq 2000). Single-nucleotide variants (SNVs) and insertions/deletions (indels) were identified (Human Genome build GRCh37, bwa-mem, Genome Analysis Toolkit HaplotypeCaller). Genetic alterations with a frequency greater than 1% in the 1000 Genomes Project database (1000 Genomes Project, National Human Genome Research Institute, www.1000genomes.org) or the ExAC database (Huh *et al*, 2009; Karczewski *et al*, 2017) were initially excluded. The minor allele frequency (MAF) of each mutation in the study group was compared to that in the general population (combined MAF across all ethnic groups, ExAC database), and statistical enrichment among iNPH patients was calculated using the two-tailed *chi-square* test with and without the Yates correction. Four publicly available computer prediction algorithms (SIFT, Provean, Mutation Tester, and Polyphen 2) were used to predict the effect of each mutation on protein function. Genes with three or more mutations that were predicted to be damaging by at least two of the four computer prediction algorithms were selected for further study. Further examination of *CWH43*, a gene that harbored the most recurrent damaging mutation, revealed another recurrent damaging deletion with a MAF of 0.0142 in the general population. This deletion resulted in a frameshift and truncation of the encoded protein and was thus included in the study. Genetic alterations were confirmed using polymerase chain reaction (PCR) and Sanger sequencing.

## Immunohistochemical staining

*In situ* mRNA hybridization images for *CWH43* were obtained from a public database (Allen Brain Atlas; Lein *et al*, 2007; Kahle *et al*, 2016). Cryostat sections of mouse brains were prepared and stained for fluorescence immunohistochemistry using antibodies directed against Cwh43 (1:500, Sigma, Human Protein Atlas HPA042814), CD59 (1:200, Santa Cruz, clone H-7), ZO-1 (1:250, Thermo Fisher, 339194), or acetylated α-tubulin (1:500, Cell Signaling, 5335S).

Nuclei were counterstained using 4′,6-diamidino-2-phenylindole (DAPI).

Ventricular ependymal cells from newborn wild-type mice were dissociated and cultured in medium containing 2% serum. The cells were then fixed in paraformaldehyde, stained using antibodies against Cwh43 and acetylated α-tubulin (to visualize cilia), and imaged using fluorescence confocal microscopy.

**Functional analysis of Cwh43 *in vitro***

Human HeLa cells harboring the *CWH43* (4:49034669 CA/C; Leu533Ter) were generated using the CRISPR/Cas9 method. The mutation was confirmed by DNA sequencing, and loss of Cwh43 protein expression was confirmed by Western blot. Expression plasmids for wild-type human *CWH43* and human *CWH43* harboring either the iNPH-associated deletion (4:49063892 CA/C; Lys696Asnf-sTer23) or (4:49034669 CA/C; Leu533Ter) were generated using site-specific mutagenesis. Expression plasmids encoding Green Fluorescent Protein (GFP) fused to the N-terminus of these *CWH43* variants were also generated. HeLa cell lines stably expressing each fusion protein were then generated by transient transfection followed by antibiotic selection. Expression plasmids encoding RFP-CD59 or RFP-Folate receptor alpha were generated and overexpressed in HeLa cells via transient transfection. To label the endoplasmic reticulum (ER), a baculovirus encoding RFP-calreticulin-KDEL (Thermo Fisher, Cell Light BacMam 2.0) was used. To label the Golgi apparatus, HeLa cells were fixed in paraformaldehyde and immunostained using an anti-golgin-97 antibody (Cell Signaling, clone D8P2K). Cell surface CD59 in HeLa cells was detected by flow cytometry using a CD59-FITC antibody (BD Pharmingen, 556640).

**Generation and analysis of CWH43 mutant mice**

C57bl6 mice harboring a Met533Ter mutation (coinciding to the human *CWH43* deletion 4:49034669 CA/C; Leu533Ter) were generated using CRISPR/Cas9 and bred to generate heterozygous ($CWH43^{WT/M533}$) and homozygous ($CWH43^{M533/M533}$) animals. To control for CRISPR/Cas9 off-target effects, we independently generated a second C57bl6 mouse line harboring one mutant Met533Ter allele and one allele containing a 10 bp deletion (ACCAGCCATA) in *CWH43* that generates a stop codon at A530 ($CWH43^{M533/A530}$). Both *CWH43* mutant mouse lines were studied for comparison.

Axial T2-weighted magnetic resonance (MR) brain images from 8 iNPH patients harboring *CWH43* deletions were obtained from medical records. Axial T2-weighted MR brain images from 54 age- and gender-matched asymptomatic individuals were obtained from a publicly available database (www.oasis-brains.org; OASIS-3: Longitudinal Neuroimaging, Clinical, and Cognitive Dataset for Normal Aging and Alzheimer's Disease). Using NIH Image J software, the cross-sectional areas of the brain and ventricles at a level just above the interthalamic adhesion were measured. The ratio of ventricular area to brain area was obtained for each case. Ratios from iNPH patients harboring *CWH43* deletions were then compared to those from asymptomatic controls. Statistical significance was calculated using the two-tailed *t*-test.

T2-weighted MR images of the brains of *CWH43* mutant mice were obtained, and ventricular volume was calculated using Image J and a custom automated computer algorithm. In some cases,

**The paper explained**

**Problem**
Idiopathic normal pressure hydrocephalus (iNPH) is a neurological disorder of aging that is characterized by enlarged cerebral ventricles, gait difficulty, incontinence, and cognitive decline. The cause and pathophysiology of iNPH are unknown. Although iNPH affects an estimated 700,000 Americans, most remain undiagnosed because many practitioners are unfamiliar with the disease or doubt its existence.

**Results**
We performed whole exome sequencing of DNA obtained from 53 unrelated iNPH patients. Damaging SNVs and indels affecting the most frequently altered gene were studied further using genetically engineered mice and human cell lines. A heterozygous damaging deletion in *CWH43* was observed in four iNPH patients and was enriched 6.6-fold among iNPH patients when compared to the general population ($P < 0.0002$, $X^2$ test). A second heterozygous damaging *CWH43* deletion was identified in four additional patients and was enriched 2.7-fold. We found that Cwh43 regulates the membrane localization of GPI-anchored proteins in mammalian cells, and both of the iNPH-associated *CWH43* deletions disrupt this function. In the mouse brain, Cwh43 expression is high in ciliated ependymal and choroid plexus cells. Mice heterozygous for *CWH43* deletions appeared grossly normal but displayed enlarged ventricles, gait and balance abnormalities, decreased numbers of ependymal cilia, and decreased localization of GPI-anchored proteins to the apical surfaces of choroid plexus and ependymal cells.

**Impact**
Approximately 15% of patients with iNPH harbor heterozygous loss of function deletions in *CWH43*. Mice harboring iNPH-associated *CWH43* deletions develop communicating hydrocephalus, gait dysfunction, and choroid plexus and ependymal cell abnormalities. Our findings provide novel mechanistic insights into the origins of iNPH and demonstrate that it represents a distinct disease entity.

mouse brains were harvested and the ventricular surface was fixed for examination using immunohistochemistry or scanning electron microscopy. Color-coded spatial analysis of gait in mice was performed using a placemat and quantitative measurement of stride length, stance, and sway. Evaluation of strength, balance, and coordination in mice was performed using the rotarod performance test. Statistical significance for laboratory studies was calculated using the two-tailed unpaired *t*-test with a significance threshold of $P < 0.05$.

**Statistical analysis**

Genetic alterations of greater than 1% in publicly available databases (1000 Genomes Project database, National Human Genome Research Institute, www.1000genomes.org or the ExAC database) were initially excluded. The minor allele frequency (MAF) of each mutation in the study group was compared to that in the general population (combined MAF across all ethnic groups, ExAC database), and statistical enrichment among iNPH patients was calculated using the two-tailed *chi-square* test with and without the Yates correction. Animals were grouped based on their genotype, mean and standard deviation were calculated for each group, and the appropriate *t*-test was used to determine the statistical significance.

## Data availability

The whole exome sequencing data from normal pressure hydrocephalus patients are available in dbGaP https://www.ncbi.nlm.nih.gov/projects/gapprev/gap/cgi-bin/study.cgi?study_id=phs002296.v1.p1). The accession ID is phs002296.v1.p1.

**Expanded View** for this article is available online.

## Acknowledgements

This work was made possible by a generous gift from Susan and Frederick Sontag. Additional support was provided by NIH R01 NS106985 and NIH R56 NS100511 from the National Institute of Neurological Disorders and Stroke to MDJ.

## Author contributions

HWY and MDJ were responsible for the design of the work. SL and PJP were responsible for the bioinformatic analysis of exome sequencing data. HWY, DY, HD, YZ, LH, SZhao, SZhan, YM, MFJ, AKR, TAJ, GW, SZheng, RSC, and MDJ were responsible for the collection of patients and experimental data. HWY, SL, RSC, PJP, and MDJ were responsible for data analysis and interpretation. HWY, RSC, and MDJ were responsible for critical review and writing of the manuscript. All authors read and approved the final manuscript.

## Conflict of interest

The authors declare that they have no conflict of interest.

## For more information

OASIS: Longitudinal: https://doi.org/10.1162/jocn.2009.21407

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
