## [Review Process File · EMBO Molecular Medicine]

Deletions in *CWH43* Cause Idiopathic Normal Pressure Hydrocephalus

Hong Wei Yang, Semin Lee, Dejun Yang, Huijun Dai, Yan Zhang, Lei Han, Sijun Zhao, Shuo Zhang, Yan Ma, Marciana Johnson, Anna Ratray, Tatyana Johnson, George Wang, Shaokuan Zheng, Rona Carroll, Peter Park, and Mark Johnson

DOI: [10.15252/emmm.202013249](https://doi.org/10.15252/emmm.202013249)

Corresponding author: Mark Johnson (mark.johnson3@umassmemorial.org)

Review Timeline:

Submission Date:	6th Aug 20
Editorial Decision:	8th Sep 20
Revision Received:	3rd Nov 20
Editorial Decision:	17th Nov 20
Revision Received:	4th Dec 20
Accepted:	10th Dec 20

Editor: Zeljko Durdevic

Transaction Report:

8th Sep 2020

Dear Prof. Johnson,

Thank you for the submission of your manuscript to EMBO Molecular Medicine. We have now received feedback from the three reviewers who agreed to evaluate your manuscript. As you will see from the reports below, all three referees are overall supportive of the study but also raise some concerns that should be addressed in a revision of the present manuscript. Particular attention should be given to co-staining of GFP-Cw43-k696fs with ER and Golgi markers in HeLa cells as well as to providing high-quality immunostaining images as suggested by the referee #1. No further in vivo experiments are required. However, addressing the reviewers' concerns in full, experimentally or in writing, will be necessary for consideration of your manuscript in our journal. Particularly, you should provide a detailed response to the referee #1 suggestions for the in vivo experiments and discuss the limitations of the study in that regard. Please be aware that the acceptance of the manuscript might entail second round of review depending on the extent and elaborateness of the revision.

EMBO Molecular Medicine encourages a single round of revision only and therefore, acceptance or rejection of the manuscript will depend on the completeness of your responses included in the next, final version of the manuscript. For this reason, and to save you from any frustrations in the end, I would strongly advise against returning an incomplete revision.

We realize that the current situation is exceptional on the account of the COVID-19/SARS-CoV-2 pandemic. Therefore, please let us know if you need more than three months to revise the manuscript.

I look forward to receiving your revised manuscript.

***** Reviewer's comments *****

Referee #1 (Remarks for Author):

In this manuscript, Yang et al. conducted a genomic analysis on 53 cases of idiopathic normal

pressure hydrocephalus (iNPH), and identified a mutation in CWH43, which is a gene encoding the transmembrane protein involved in ceramide transfer. Subsequently, they characterized the subcellular-localization, tissue expression, and their pathological effects on the brain ventricle as well as ependymal and choroid plexus epithelial cilia. Collectively, they concluded that this mutation is one of the candidate gene for iNPH that can recapitulate the phenotype in model animal, and this may be a toehold for dissecting the molecular mechanisms responsible for the pathogenesis of iNPH.

This Reviewer find the framework of story coherent and intriguing enough to attract the interests of readership in EMBO Molecular Medicine. However, there are several points, especially on the quality of data, which have to be addressed before making a final decision.

(specific critique)

1. Regarding the localization of GFP-Cw43-k696fs, fig.1C), this Reviewer cannot find the panel showing explicitly the reduced localization of this mutant fusion protein, as they did not conduct a co-staining with anti-golgin97. While the authors stated that "decreased association with the Golgi apparatus" in figure caption, this has to be verified by real data.
2. The quality of immunostaining seems to be generally below the standard. For example, the panels in fig 2B, both the staining with anti-ac-tubulin and Cwh43 are blurred and ambiguous. If this intended to represent the ependymal cilia, the staining here cannot witness the exact structure. They have to be replaced with clearer one or re-visited by confocal microscopy. As for Appendix figS8C, the staining with anti-ac-tubulin is saturated that has to be easily remedies by re-trial by diluting the first antibody.
3. This Reviewer just wonders why the authors employed different color of second antibody for the middle panel (M533/M533) in fig4C. Moreover. While have quantified the number of cilia in ependymal cells (panelA), the immunostaining in panelC was made on the choroid plexus epithelium. I think figS9 would be preferrable to be placed here as it is focused on the ependyma.
4. Discussion section is mostly devoted to the literature survey. More detailed evaluation of experimental data in the light of previous study are required, especially when it comes to the cell biological significance of Cwh43 with references to the pathogenesis of iNPH.
5. What about the motility of ependymal cilia in mutant mice? It would be nice to assess the motility and flow of ependymal cilia ex vivo (Nonami et al., 2013, Cytoskeleton).
6. If possible, analysis of mutant mice older than 1 year would be ideal to support the authors' conclusion and be concordant with the clinical course of human patients.
7. Minor arguments
 - Page3, line4, What does it intend to say? Americans (4)
 - Page4, line1, Citation for CFAP43 is required.
 - Page7, line7, Gross appearance of whole brain and the evidence for the communicating hydrocephalus would be preferably shown as a Supplemental.
 - Page7, third paragraph, Footprints of gaiting of mutant and WT mice are better presented in the Supplemental.
 - Page7, the last sentence, Regarding the localization of CD59 on primary cilia, previous study reported on RPE1 cells (Madugula and Lu, J. Cell Sci., 129:3922-3934, 2016). This may strengthen the authors' conclusion.

Referee #2 (Comments on Novelty/Model System for Author):

This is one of the most important studies in the field of iNPH research so far, i.e., has potential to be

a groundbreaker.

Referee #2 (Remarks for Author):

This paper by Yang et al. report potential association of two loss of function deletions in CWH43 gene with idiopathic normal pressure hydrocephalus (iNPH). The finding is novel and the relevance is confirmed by extensive functional analysis. This is one of the most important studies in the field of iNPH research so far, i.e., has potential to be a groundbreaker.

The major questions:

The discovery cohort(s) is rather small but did the authors find any other potential loci related with iNPH? In addition, any other iNPH cohorts to replicate the finding?

Minor notes:

I suggest to omit "sporadic" from the topic since 3 out of the 8 patients with CWH43 alteration have potential family history related with iNPH. This by no means dilute the novelty of the paper.

In abstract and introduction I suggest to change term "dementia" (as a symptom) to "cognitive decline" or analogous.

Page 4, two last paragraphs: please check MAFs of CWH43 deletions; is 0.0377 correct in both locations?

Discussion, page 8, 2nd para, last sentence: How is the "...consistent with an autosomal dominant pattern of inheritance" related with the first part of the clause?

Materials and Methods, page 10:

Were all the three separate patient cohorts recruited from the same institution, i.e., what makes them discrete?

It seems that the "general population" is from ExAC database? Please describe that little more detail like the number subjects and age range as well as the reference. Where does "43" refer?

Figure 3: Please check the letters; I guess that B in the upper should be C.

Referee #3 (Comments on Novelty/Model System for Author):

iNPH is a dementia disease, overlapping with Alzheimer's. Revealing the genetic background for dementia subtypes is of significance and of interest to scientists/clinicians.

Referee #3 (Remarks for Author):

This is a study about genetic alterations in sporadic idiopathic normal pressure hydrocephalus, demonstrating deletion of CWH43. Moreover, CWH43 deletion in mice induced ventriculomegaly and gait ataxia. The study is novel and of definitive interest. I have some minor comments.

The CWH43 deletion was observed in 4+4 of 53 iNPH patients. Though statistical significant as compared to the general population, the number is small. This should be commented on.

Three of 8 iNPH patients with CWH43 deletion had familial iNPH. Is it then correct to refer to these

as sporadic iNPH? At least it should be commented on. Was the result significant when only including those without family members with iNPH?. Further, given that 3/8 had familiar iNPH, I feel "sporadic" should be removed from the title.

There are few papers on genetics of sporadic iNPH. The largest series reported on is by Korhonen et al *Neurol Genet*, vol 4 issue 6, pp e291, 2018. This should be referred to.

***** Reviewer's comments *****

Referee #1 (Remarks for Author):

In this manuscript, Yang et al. conducted a genomic analysis on 53 cases of idiopathic normal pressure hydrocephalus (iNPH), and identified a mutation in CWH43, which is a gene encoding the transmembrane protein involved in ceramide transfer. Subsequently, they characterized the subcellular-localization, tissue expression, and their pathological effects on the brain ventricle as well as ependymal and choroid plexus epithelial cilia. Collectively, they concluded that this mutation is one of the candidate gene for iNPH that can recapitulate the phenotype in model animal, and this may be a toehold for dissecting the molecular mechanisms responsible for the pathogenesis of iNPH.

This Reviewer find the framework of story coherent and intriguing enough to attract the interests of readership in EMBO Molecular Medicine. However, there are several points, especially on the quality of data, which have to be addressed before making a final decision.

(specific critique)

1. Regarding the localization of GFP-Cw43-k696fs, fig.1C), this Reviewer cannot find the panel showing explicitly the reduced localization of this mutant fusion protein, as they did not conduct a co-staining with anti-golgin97. While the authors stated that "decreased association with the Golgi apparatus" in figure caption, this has to be verified by real data.

As requested by the reviewer, we have added panels to Figure 1C demonstrating co-staining using anti-golgin97 with mutant GFP-Cwh43-k696fs and mutant GFP-Cwh43-Leu533ter that demonstrates decreased association of these mutants with the Golgi apparatus.

2. The quality of immunostaining seems to be generally below the standard. For example, the panels in fig 2B, both the staining with anti-ac-tubulin and Cwh43 are blurred and ambiguous. If this intended to represent the ependymal cilia, the staining here cannot witness the exact structure. They have to be replaced with clearer one or re-visited by confocal microscopy. As for Appendix figS8C, the staining with anti-ac-tubulin is saturated that has to be easily remedies by re-trial by diluting the first antibody.

We apologize for the quality of some of the immunostaining included in the previous version. As suggested, we have provided better quality images which were taken on a higher magnification confocal microscope. These images clearly demonstrate immunoreactivity for Cwh43 and acetylated tubulin (cilia marker) in the ependymal layer. We have reduced the exposure and contrast in the images in Appendix Fig. S8C (now Fig. S9C) to decrease the saturation of the acetylated tubulin staining as suggested.

3. This Reviewer just wonders why the authors employed different color of second antibody for the middle panel (M533/M533) in fig4C. Moreover. While have quantified the number of cilia in ependymal cells (panelA), the immunostaining in panelC was made on the choroid plexus epithelium. I think figS9 would be preferable to be placed here as it is focused on the ependyma.

The staining in the micrographs performed in Fig. 4C was performed at separate times and a different color antibody was used each time. As requested by the reviewer, we have moved the data from Fig. S9 to Fig 4C showing a loss of apical localization of CD59 in the ependymal cells of CWH43 mutant mice.

4. Discussion section is mostly devoted to the literature survey. More detailed evaluation of experimental data in the light of previous study are required, especially when it comes to the cell biological significance of Cwh43 with references to the pathogenesis of iNPH.

In response to the reviewer's concerns, we have added the following paragraph which provides a more detailed evaluation of the experimental data, and more clearly discusses the biological significance of CWH43 in the context of the pathogenesis of iNPH.

“In yeast, the incorporation of ceramide into the lipid anchor of GPI-anchored proteins by Cwh43 promotes the retention of these proteins on the plasma membrane (Yoko et al, 2018). Loss of Cwh43 function in yeast leads to aberrant expression of these proteins on the cell wall. The Golgi apparatus is thought to be a primary site where the lipid anchor of GPI-anchored proteins is modified and where GPI-anchored proteins are incorporated into vesicles for transport to their final destination on the cell membrane (Matheson et al, 2006). We observed strong expression of Cwh43 in the Golgi apparatus of mammalian cells. However, both of the iNPH-associated *CWH43* deletions cause a loss of Cwh43 protein in the Golgi apparatus and disrupt the targeting of certain GPI-anchored proteins (e.g. CD59) to lipid microdomains in the plasma membrane and to the apical membrane of polarized ventricular epithelial cells. Our data thus provide the first evidence that Cwh43 regulates the subcellular targeting of GPI-anchored proteins in mammalian cells. We hypothesize that mislocalization of GPI-anchored proteins in choroid plexus and ependymal epithelial cells harboring mutant Cwh43 disrupts the normal function of these cells.”

5. What about the motility of ependymal cilia in mutant mice? It would be nice to assess the motility and flow of ependymal cilia ex vivo (Nonami et al., 2013, Cytoskeleton).

We agree with the reviewer that this is an interesting question. We have begun experiments to address this question. Our preliminary data suggest there is no difference in cilia beat frequency between wild type and *CWH43* mutant mice. These studies are ongoing and will be reported in a subsequent manuscript.

6. If possible, analysis of mutant mice older than 1 year would be ideal to support the authors' conclusion and be concordant with the clinical course of human patients.

We attempted to address this question in older mice. We again agree with the reviewer that studying older mice would be ideal. We have used serial MRI to follow wild type and *CWH43* mutant mice over time and found that the ventricles in both of the cohorts increased with age. There was a trend for the *CWH43* mutant mice to have larger ventricles than wild type mice at 18 months of age, but a reduction in the size of the cohorts due to the age-related deaths of some of the animals (both wild type and mutant) during the prolonged course of this experiment prevented us from performing this study as intended.

7. Minor arguments

- Page3, line4, What does it intend to say? Americans (4)

We have edited this sentence to read as follows:

“According to the Hydrocephalus Association, iNPH affects about 700,000 Americans.”

- Page4, line1, Citation for *CFAP43* is required.

We have added the reference as requested. The sentence now reads as follows:

“One study identified a CFAP43 mutation as a possible cause of familial iNPH (Morimoto et al, 2019).”

• Page7, line7, *Gross appearance of whole brain and the evidence for the communicating hydrocephalus would be preferably shown as a Supplemental.*

As requested by the reviewer, we have added images to the supplemental data (Appendix Figure S7) showing a patent aqueduct and flow of 70 KD fluorescent dextran from the lateral ventricles through the aqueduct and into the paravascular spaces of the cortex in homozygous *CWH43* mutant mice. We have added a figure legend that reads as follows:

Appendix Figure S7. *CWH43* mutant mice develop communicating hydrocephalus.

A) Images of the whole brain obtained from *CWH43*^{WT/WT}, *CWH43*^{WT/M533}, and *CWH43*^{M533/M533} mice.
B) Fluorescence micrographs of cryostat sections of a brain from a *CWH43*^{M533/M533} mouse obtained at the level of cerebral aqueduct and periaqueductal gray (arrow). The brain sections have been stained with DAPI to identify cell nuclei (blue). Prior to harvesting the brain, the lateral ventricle was injected with 5 µl of saline containing a fluorescent dextran (70 KD). After 5 minutes, the brain was harvested and processed for fluorescence microscopy. Fluorescent dextran completely filled the ventricular system and paravascular spaces in the cortex, confirming patency of the cerebral aqueduct. Scale is approximately 5 mm.

• Page7, third paragraph, *Footprints of gaiting of mutant and WT mice are better presented in the Supplemental.*

We would prefer to keep the footprints of the gait results in the main text. Gait and balance difficulties are the most common symptoms of iNPH. As such, their presence here provides strong support that this animal model phenocopies iNPH. The importance of this finding thus merits its location in the main text.

• Page7, the last sentence, *Regarding the localization of CD59 on primary cilia, previous study reported on RPE1 cells (Madugula and Lu, J. Cell Sci., 129:3922-3934, 2016). This may strengthen the authors' conclusion.*

We have examined the report by Madugula and Lu. They discuss the presence of sequences in CD8 that target the protein to cilia. CD8 is a transmembrane protein and not a GPI-anchored protein.

Referee #2 (Comments on Novelty/Model System for Author):

This is one of the most important studies in the field of iNPH research so far, i.e., has potential to be a groundbreaker.

Referee #2 (Remarks for Author):

*This paper by Yang et al. report potential association of two loss of function deletions in *CWH43* gene with idiopathic normal pressure hydrocephalus (iNPH). The finding is novel and the relevance is confirmed by extensive functional analysis. This is one of the most important studies in the field of iNPH research so far, i.e., has potential to be a groundbreaker.*

The major questions:

The discovery cohort(s) is rather small but did the authors find any other potential loci related with iNPH? In addition, any other iNPH cohorts to replicate the finding?

We have indeed identified other loci that appear to be statistically associated with iNPH. We allude to the possible existence of such loci in the discussion of the current manuscript. We are currently working to perform validation experiments for these loci at this time. We have begun to collect DNA from a larger cohort of iNPH patients with the goal of further replicating these findings (beyond the 3 cohorts reported here) and identifying additional loci.

Minor notes:

I suggest to omit "sporadic" from the topic since 3 out of the 8 patients with CWH43 alteration have potential family history related with iNPH. This by no means dilute the novelty of the paper.

In response to the reviewer's suggestions, we have removed the term sporadic from the manuscript.

In abstract and introduction I suggest to change term "dementia" (as a symptom) to "cognitive decline" or analogous.

In response to the reviewer's suggestion, we have changed the term dementia to "cognitive decline" in the abstract and introduction.

Page 4, two last paragraphs: please check MAFs of CWH43 deletions; is 0.0377 correct in both locations?

This is correct, both deletions were found in 4 of the 53 patients (4/106 alleles = .0377).

Discussion, page 8, 2nd para, last sentence: How is the "...consistent with an autosomal dominant pattern of inheritance" related with the first part of the clause?

When a single defective allele can lead to disease (i.e. heterozygous, as mentioned at the beginning of the sentence), the pattern of inheritance that is observed is autosomal dominant. By contrast, when both alleles must be affected before disease occurs, the pattern of inheritance that is seen is autosomal recessive. This latter pattern is seen for most cases of congenital hydrocephalus that are genetic in origin and, consequently, such cases are quite rare when compared to the prevalence of iNPH.

Materials and Methods, page 10:

Were all the three separate patient cohorts recruited from the same institution, i.e., what makes them discrete?

All three separate patient cohorts were recruited from the same institution. They are discrete because they were recruited and analyzed at three separate times in a sequential manner over a period of years. In other words, the first cohort was recruited and the DNA was isolated, sequenced and analyzed to identify iNPH-associated loci. Then, a second cohort was recruited and the DNA was again isolated, sequenced and analyzed to identify iNPH-associated loci. Finally, a third cohort was recruited and the DNA was isolated, sequenced and analyzed. CWH43 deletions were identified at a rate significantly greater than expected in all 3 cohorts.

It seems that the "general population" is from ExAC database? Please describe that little more detail like the number subjects and age range as well as the reference. Where does "43" refer?

The number 43 was a reference from an earlier version of the manuscript and has been removed. The ExAC database contains exomes from 60,706 individuals from around the world. The data is presented in terms of race/ethnicity. The reference that describes the database is:

Lek M et al. Analysis of protein-coding genetic variation in 60,706 humans. *Nature* 2016; 536:285-91.

Figure 3: Please check the letters; I guess that B in the upper should be C.

We thank the reviewer for pointing this out to us. We have corrected the error and labeled the panels correctly in the revised figures.

Referee #3 (Comments on Novelty/Model System for Author):

iNPH is a dementia disease, overlapping with Alzheimer's. Revealing the genetic background for dementia subtypes is of significance and of interest to scientists/clinicians.

Referee #3 (Remarks for Author):

This is a study about genetic alterations in sporadic idiopathic normal pressure hydrocephalus, demonstrating deletion of CWH43. Moreover, CWH43 deletion in mice induced ventriculomegaly and gait ataxia. The study is novel and of definitive interest. I have some minor comments. The CWH43 deletion was observed in 4+4 of 53 iNPH patients. Though statistically significant as compared to the general population, the number is small. This should be commented on.

We have added the following statement to the Discussion:

“Although each of the three cohorts in our study was limited in size, we nevertheless observed that *CWH43* deletions were overrepresented in each one.”

Three of 8 iNPH patients with CWH43 deletion had familial iNPH. Is it then correct to refer to these as sporadic iNPH? At least it should be commented on. Was the result significant when only including those without family members with iNPH? Further, given that 3/8 had familial iNPH, I feel "sporadic" should be removed from the title.

*There are few papers on genetics of sporadic iNPH. The largest series reported on is by Korhonen et al *Neurol Genet*, vol 4 issue 6, pp e291, 2018. This should be referred to.*

Although all of the patients with *CWH43* deletions in this study initially presented as if they were sporadic, careful questioning revealed either a relative with diagnosed iNPH or symptoms suggestive of possible iNPH in 3 such patients. We agree with the reviewer that, once family members have been identified who have been diagnosed with iNPH, these cases cannot technically be called “sporadic”. We have therefore removed the term from the manuscript. Also, in response to the reviewer’s suggestion, we have added a sentence and a reference regarding the large iNPH genotyping study of the *SFMBT1* gene by Korhonen et al. The sentence reads as follows:

“Another large multinational study of Finnish and Norwegian iNPH patients used genotyping of *SFMBT1* to identify copy number loss in intron two of *SFMBT1* in 10% of Finnish (odds ratio [OR] = 1.9, $p = 0.0078$) and 21% of Norwegian (OR = 4.7, $p < 0.0001$) patients with iNPH (Korhonen et al, 2018), although the significance of this finding is yet to be determined.”

17th Nov 2020

Dear Prof. Johnson,

Thank you for the submission of your revised manuscript to EMBO Molecular Medicine. I am pleased to inform you that we will be able to accept your manuscript pending the following final amendments:

1) In the main manuscript file, please do the following:

- Correct/answer the track changes suggested by our data editors by working from the attached/uploaded document.

***** Reviewer's comments *****

Referee #1 (Comments on Novelty/Model System for Author):

The quality of data has been improved to meet the requirements of this reviewer. This reviewer finds the study important for the community of this field.

Referee #1 (Remarks for Author):

All the primary concerns of this reviewer have been fully addressed in the revised manuscript. This reviewer recommends the publication of this manuscript in EMBO Molecular Medicine.

The authors performed the requested changes.

10th Dec 2020

Dear Prof. Johnson,

We are pleased to inform you that your manuscript is accepted for publication.

Corresponding Author Name: Mark D. Johnson

Manuscript Number: EMM-2020-13249